# Energy Saving in Permanent Cardiac Pacing: Pulse Waveform and Charge Balancing Deserve Consideration

**DOI:** 10.3390/bioengineering12020194

**Published:** 2025-02-17

**Authors:** Franco Di Gregorio, Lina Marcantoni, Aldo Mozzi, Alberto Barbetta, Francesco Zanon

**Affiliations:** 1Clinical Research Unit, Medico Spa, 35030 Rubano, PD, Italy; fgd.digregorio@alice.it (F.D.G.); barbetta@medicoweb.com (A.B.); 2Cardiology Department, Santa Maria della Misericordia GH, 45100 Rovigo, RO, Italy; franc.zanon@iol.it; 3R&D Department, Medico Spa, 35030 Rubano, PD, Italy; aldo_mozzi@yahoo.it

**Keywords:** rheobase, chronaxie, safety margin, pacing energy, optimal pulse duration, useful pulse duration

## Abstract

The pacing pulse produced by implantable stimulators can be described as a truncated exponential decay from the starting peak amplitude, corresponding to the discharge of the output stage capacitance (reservoir and isolation capacitors, in series) along the application time. Pulse decay and charge balancing have relevant implications on the ideal setting of a pacing device, as demonstrated by mathematical predictions based on well-acknowledged theoretical statements. Successful stimulation is achieved with minimum energy expense at a pulse duration shorter than the chronaxie time, which represents the upper border of the advisable duration interval. With any start amplitude, the stimulation safety margin can be improved by a duration increase beyond the chronaxie only up to an absolute limit (longest useful duration), which depends on the chronaxie and the pulse time-constant. At the longest useful duration, the threshold start amplitude is at the minimum and cannot decrease any further, though it and the corresponding pulse mean amplitude largely exceed the rheobase. The overall pacing performance is affected, in addition, by the load resistance and the electrode capacitance. Pulse amplitude decay limits the effectiveness of extended duration in implantable stimulators, making short pulses preferable whenever possible. Proper pulse settings based on actual waveform properties can prevent energy waste and reduce pacing consumption, thus prolonging the service life of the stimulator.

## 1. Introduction

In all of its applications, ranging from restoration of cardiac rhythm and atrio-ventricular conduction to cardiac resynchronization, the essential aim of cardiac pacing is to release a properly timed electric pulse suitable to depolarize the target cell membrane to a critical value, at which voltage-gated sodium channels are opened to trigger action potential generation and conduction [1,2]. As an implanted pacing device must rely on a limited energy source, effective stimulation should be better achieved with minimum battery consumption. Continuous progress in the implemented technology, stimulator design, and materials employed in pacing leads and electrodes have allowed substantial advancement toward this goal. Nevertheless, an appropriate location of the pacing electrode as close as possible to the target tissue, as well as careful management of the implanted pacing system, are still mandatory.

The evolution of increasingly sophisticated autoregulation algorithms has reduced the need for manual programming of the stimulator pulse amplitude. In contrast, choosing the appropriate pulse duration is still a user’s task, which can have relevant implications on the stimulation efficiency and energy cost. Some principles of biophysics and cellular electrophysiology should be better applied for this purpose. They are reviewed in the present paper, devoting special attention to the correct interpretation of the classic excitation theory when stimulation is performed by implantable pacemakers, which produce electric pulses of declining amplitude. Moreover, the stimulator consumption depends on the energy released in both pulse emission and related charge balancing (i.e., post-spike discharge of the device isolation capacitors). The actual pulse waveform and the impact of charge balancing should be considered to properly set any pacing device featuring an output stage with limited overall series capacitance [3,4]. The difference with respect to widely used stimulation models, focusing just on the pulse energy and often assuming a squared pulse waveform, has been derived mathematically and further demonstrated by numerical simulation, assuming common conditions in conventional myocardial pacing. Ready-to-use quantitative solutions to issues frequently met in the clinical setting, yet often ignored or empirically managed, are provided to foster the conscious utilization of battery charge, which can contribute to prolonging the device’s service life without reducing (or even increasing) the actual safety of permanent pacing.

## 2. Lapicque’s Law of Excitation

According to Lapicque, the response of any excitable tissue undergoing electrical stimulation is precisely predicted by two parameters, rheobase and chronaxie, whose values can be different in different implants, depending on substrate excitability, performance of the stimulating system, distance between pacing electrode and target, and electrical properties of the tissues in between. The threshold strength (i.e., the minimum pulse amplitude required for effective membrane depolarization) decreases as a function of pulse duration, according to the following equations [3,4,5,6,7]:(1)$$I_{m\_th}(t_{1})=I_{r}\left(1+\frac{t_{c}}{t_{1}}\right)$$(2)$$V_{m\_th}(t_{1})={R\;I}_{m\_th}={R\;I}_{r}\left(1+\frac{t_{c}}{t_{1}}\right)$$
where $I_{m\_th}$ is the mean current intensity of the threshold pulse as a function of pulse duration $\left(t_{1}\right)$;$\;V_{m\_th}$ is the corresponding mean voltage according to Ohm’s law, assuming a purely resistive load (i.e., negligible electrode polarization); R is the load resistance (approximated by the pacing impedance measured by the device); $I_{r}$ is the rheobase current (current intensity of the threshold pulse of infinite duration); $t_{c}$ is the chronaxie time (threshold duration of a pulse with mean amplitude equal to twice the rheobase). The relation between the threshold mean current or voltage versus pulse duration is referred to as the “Lapicque curve” in the following (Figure 1a, solid curve).

By multiplying both sides of Equation (1) by the pulse duration $\left(t_{1}\right)$:(3)$${Q_{p\_th}\left(t_{1}\right)=I}_{m\_th}\;t_{1}=I_{r}\left(1+\frac{t_{c}}{t_{1}}\right)\;t_{1}=I_{r}\;t_{1}+I_{r}\;t_{c}$$
the product of mean threshold current and application time, that is, the charge of the threshold pulse $\left(Q_{p\_th}\right)$, is described as a linear function of pulse duration, with slope equal to the rheobase current $\left(I_{r}\right)$ and intercept equal to the product of rheobase current and chronaxie [5,6,7]. As the slope is positive, the threshold charge increases with increasing pulse duration, and the intercept represents the smallest amount of charge needed to excite the target $\left(Q_{p\_th\_min}\right)$ with pulse duration approaching 0 (Figure 1b, thick solid line). Therefore,(4)$$t_{c}=\frac{Q_{p\_th\_min}}{I_{r}}$$(5)$$V_{r}={R\;I}_{r}$$
where $V_{r}$ is the rheobase potential (voltage of the threshold pulse of infinite duration). If the threshold charge is determined at various pulse durations, rheobase and chronaxie can be derived by the linear regression of the experimental data. Once the two excitability constants are defined, the mean amplitude of the threshold pulse can be predicted for any pulse duration and expressed as either current (Equation (1)) or voltage (Equation (2)). This is useful for working out the pulse parameters allowing effective pacing with minimum energy consumption, as shown in the following.

Rearranging Equation (3) yields a definition of the stimulation threshold where either the mean current or the pulse duration can be assumed to be the dependent variable ($I_{m\_th}{(t}_{1})$ or $t_{1\_th}\left(I_{m}\right),$ respectively):(6)$${t_{1}\;(I}_{m\_th}-I_{r})={t_{1\_th}\;(I}_{m}-I_{r})=I_{r}\;t_{c}=Q_{p\_th\_min}$$

In both instances, the threshold is reached when the product of pulse duration times the difference between mean and rheobase current equals the minimum threshold charge, that is, the charge allowing threshold depolarization of the cell membrane in the liminal myocardial volume (the smallest activated volume which can initiate action potential propagation). The threshold duration for any mean current intensity is thus defined as:(7)$$t_{1\_th}=\frac{I_{r}\;t_{c}}{I_{m}-I_{r}}$$

## 3. Pulse Production Mechanism and Charge Calculation

It must be stressed that Lapicque’s law defines the relation between duration and mean amplitude of the threshold pulse. In contrast, when a threshold test is performed in practice, the lowest voltage step resulting in effective stimulation (which depends on the device output resolution) is usually recorded as the best approximation of the ideal threshold. Even more important, the pulse start amplitude $V_{p}$, which is the programmable parameter generally known to the user [8,9,10], corresponds to the mean voltage only if the instantaneous potential V(t) is stable $\left(V\left(t\right)=V_{p}\right)$. This is not the case with present implantable devices, where the pacing pulse is produced by the discharge of a capacitance, with resulting decay of both voltage and current $\left(I(t)\right)$ during the pulse time-course [4,11]. The following exponential relations apply (Figure 2):(8)$$V(t){\;=V_{p}\;e}^{-t/\tau }$$(9)$$I(t)\;=\frac{V_{p}}{R}\;e^{-t/\tau }$$(10)$$\tau =R\;C_{out}$$
where $V_{p}$ is the peak voltage at pulse onset, R is the load resistance (the equivalent resistance of the lead coils, the electrodes, and the tissue, all summed in series), $V_{p}/R$ is the peak current at pulse onset, τ is the pulse time-constant (the longer τ, the slower the amplitude decline), $C_{out}$ is the capacitance of the stimulator output stage, and t is the pulse application time, hereby referred to as $t_{1}$ when it represents the total pulse duration.

The pulse mean voltage $\left(V_{m}\right)$ can be calculated as the definite integral of Equation (8) along the pulse time-course, divided by the pulse duration:(11)$$V_{m}(t_{1})=\frac{1}{t_{1}}\int_{0}^{t_{1}}{V_{p}\;e}^{-t/\tau }dt=\tau \;V_{p}\frac{\;\left(1-e^{{-t}_{1}/\tau }\right)}{t_{1}}$$

The pulse mean current $\left(I_{m}\right)$ is readily derived, since $V_{m}=R\;I_{m}$ by Ohm’s law and $\tau =R\;C_{out}$ as stated in Equation (10):(12)$$I_{m}(t_{1})=C_{out}\;V_{p}\;\frac{\;\left(1-e^{{-t}_{1}/\tau }\right)}{t_{1}}$$

The product of the mean current and duration provides the pulse charge $\left(Q_{p}\right)$:(13)$$Q_{p}\left(t_{1}\right)=I_{m}\;t_{1}{=C}_{out}\;V_{p}\;\left(1-e^{{-t}_{1}/\tau }\right)=C_{out}\;\left(V_{p}-V_{\begin{array}{c} 1 \end{array}}\right)$$
where $V_{1}$ is the voltage at the waveform trailing edge. If the mean voltage is definite, the corresponding start voltage can be obtained by rearranging Equation (11):(14)$$V_{p}(t_{1})=\frac{V_{m\;}t_{1}}{\tau \;\left(1-e^{{-t}_{1}/\tau }\right)}=\frac{I_{m\;}t_{1}}{C_{out}\;\left(1-e^{{-t}_{1}/\tau }\right)}$$

The nominal pulse amplitude, as shown by the device programmer, usually (yet not always) corresponds to the start voltage, as the mean voltage depends, in addition, on the output capacitance, the pulse duration, and the load resistance, which is unknown before the implantation. The relation between peak and mean voltage is thus reported, only in official documents, for a limited set of combinations between pulse parameters and the load. In contrast, Equations (11)–(13) allow working out the pulse mean voltage, current, and total charge in any pacing condition by using available information like the pulse start amplitude and duration and the pacing impedance, which is a suitable approximation of the load resistance.

The stimulator output capacitance is a technical specification found in the user’s manual or provided by the manufacturer upon request. It generally results from the combination of three capacitors: a “reservoir” capacitor charged to the pulse start voltage by the power supply system and two isolation capacitors, each facing either the cathode (tip electrode of the pacing lead) or the anode (ring electrode or stimulator case, in bipolar or unipolar configuration, respectively). When the pulse is over, the reservoir capacitor is disconnected, and the two isolation capacitors are short-circuited. This inverts the current flowing through the load and avoids net charge accumulation at the electrode–tissue interface, which would be detrimental on both sides (charge balancing). On the other hand, the insulation capacitors limit the equivalent capacitance of the output stage $\left(C_{out}\right)$, since they are in series with the reservoir capacitor $\left(C_{res}\right)$ during the pulse release. Therefore,(15)$$C_{out}=\frac{1}{\frac{1}{C_{res}}+\frac{1}{C_{c}}+\frac{1}{C_{a}}}$$
where $C_{c}$ and $C_{a}$ represent the capacitance of the isolation capacitor connected to the cathode or the anode, respectively. As a result, relevant voltage and current decline during the pulse course is a common feature in present implantable stimulation devices.

Unless directly assessed as the product of the output capacitance times the difference of measured leading and trailing edge voltage, the pulse charge should properly be calculated taking into consideration the time-constant of each implant, as shown in Equation (13). In daily practice, however, the amplitude decay is often ignored to simplify the task by assuming a squared waveform. In such an instance,(16)$$Q_{p{}^{\circ}}\left(t_{1}\right)=\frac{V_{p{}^{\circ}\;}t_{1}}{R}$$
$Q_{p{}^{\circ}}$ being the charge of a pulse with stable voltage, where the start and mean amplitude are equal ($V_{p{}^{\circ}}=V_{m}$). This approximation overestimates the pulse charge with respect to the more precise exponential decay model. The relative error $\left(Q_{p{}^{\circ}}/Q_{p}\right)$ increases with increasing pulse duration or decreasing time constant:(17)$$\frac{Q_{p{}^{\circ}}}{Q_{p}}=\frac{t_{1}}{\tau \;\left(1-e^{{-t}_{1}/\tau }\right)}$$

The square-wave simplification, which is never fully correct with currently used technology, thus becomes unacceptable in the case of low pacing impedance or long pulses. Furthermore, since the error depends on the pulse width, it substantially alters the assessment of rheobase and chronaxie, which relies on the relation between threshold pulse charge and duration. An example is provided in Figure 1, where the threshold start amplitude is converted into the corresponding charge by considering or ignoring a pulse decay with a 2 ms time-constant. If the decay is neglected, the threshold charge appears as a non-linear function of pulse duration, with the best linear fit showing an increased slope and reduced intercept with respect to the expectation. As a result, the corresponding rheobase current and voltage $\left(I_{r{}^{\circ}};V_{r{}^{\circ}}\right)$ are overestimated, while the chronaxie time $\left(t_{c{}^{\circ}}\right)$ is underestimated. The errors can easily represent a large fraction of the true parameters. By assuming $I_{r}=1\;mA$, $t_{c}=0.4\;ms$, and R=500 Ohm, $I_{r{}^{\circ}}$ comes out as 1.38 mA and $t_{c{}^{\circ}}$ 0.25 ms. If the load resistance decreases to 400 Ohm, $I_{r{}^{\circ}}$ further increases to 1.48 mA, and $t_{c{}^{\circ}}$ decreases to 0.23 ms.

Notwithstanding, the apparent rheobase and chronaxie based on the steady voltage assumption still provide an acceptable approximation of the threshold start amplitude as a function of pulse duration, as the errors partially compensate for each other. If setting a pacing output that ensures the required safety margin is the only aim, the square-wave model can be useful and reliable. In contrast, other applications impose the proper determination of true excitability parameters with special regard to the chronaxie. In such cases, the pulse charge must be calculated, taking into account the exponential decay in the pulse voltage and current.

## 4. Pulse Duration and Threshold Energy

Lapicque’s theory states that stimulation effectiveness and safety depend on the charge released by the pulse as a function of pulse duration. To optimize the pacing efficiency, the due pulse charge should be provided by reducing the charge drained from the battery, which is proportional to the supplied energy. The total energy expense equals the sum of the energy released by the pulse ($W_{p}$), the energy stored in the isolation capacitors ($W_{a,c}$), which is released by charge balancing after each pulse, and the extra energy spent to charge up the reservoir capacitor and regulate the stimulator output. The latter component of the overall pacing consumption depends on the technology implemented in different devices and cannot, therefore, be considered in a general analysis of common issues. Focusing on the part of pacing energy released to the load per cardiac cycle ${(W}_{l})$:(18)$$W_{l}=W_{p}+W_{a,c}\;$$
where(19)$$W_{p}\left(t_{1}\right)=\int_{0}^{t_{1}}V\;I\;dt=\frac{V_{p}^{2}}{R}\int_{0}^{t_{1}}e^{-2t/\tau }dt=\frac{V_{p}^{2}\;C_{out}}{2}\;\left(1-e^{-2t_{1}/\tau }\right)$$(20)$$W_{a,c}(t_{1})=\frac{Q_{p}^{2}}{2{\;C}_{a,c}}=\frac{{\left[C_{out}\;V_{p}\;\left(1-e^{{-t}_{1}/\tau }\right)\right]}^{2}}{2{\;C}_{a,c}}$$

The pulse charge $Q_{p}$ is defined as in Equation (13), and $C_{a,c}=C_{a}C_{c}/\left(C_{a}+C_{c}\right)$ is the equivalent capacitance of the two isolation capacitors connected in series. From the above,(21)$$W_{l}(t_{1})=\frac{V_{p}^{2}\;C_{out}}{2}\left[\left(1-e^{-2t_{1}/\tau }\right)+\frac{C_{out}{\;\left(1-e^{{-t}_{1}/\tau }\right)}^{2}}{C_{a,c}}\right]$$

If the pulse corresponds to a squared waveform directly delivered to the load, featuring constant voltage ($V_{p{}^{\circ}}$) and current ($I_{p{}^{\circ}}$), the pulse energy reduces to the following:(22)$$W_{p{}^{\circ}}\left(t_{1}\right)=V_{p{}^{\circ}}\;I_{p{}^{\circ}}\;t_{1}=\frac{V_{p{}^{\circ}}^{2}\;t_{1}}{R}$$

Since $V_{p{}^{\circ}}=V_{m}$, the energy of the threshold pulse as a function of its duration is readily obtained by inserting Equation (2) into (22):(23)$$W_{p{}^{\circ}\_th}\left(t_{1}\right)=\frac{{\left[V_{r}\;\left(t_{1}+t_{c}\right)\right]}^{2}}{R\;t_{1}}$$

The first derivative of Equation (23) is:(24)$$\frac{{dW}_{p{}^{\circ}\_th}}{dt_{1}}=\frac{V_{r}^{2}\;\left(t_{1}-t_{c}\right)\;\left(t_{1}+t_{c}\right)}{R\;t_{1}^{2}}$$
which equals 0, with a positive second derivative, if $t_{1}=t_{c}$. With squared pulses, therefore, a duration equal to the chronaxie allows threshold pacing with minimum energy expense [7].

If the pulse features exponential decay, the threshold start amplitude is found by inserting Equation (2) into (14):(25)$$V_{p\_th}(t_{1})=\frac{V_{r}\;\left(t_{1}+t_{c}\right)}{\tau \;\left(1-e^{{-t}_{1}/\tau }\right)}$$

This $V_{p}$ specification, applied in Equation (19), provides the energy of the threshold pulse (Figure 3, dashed curve):(26)$$W_{p\_th}(t_{1})={\left[\frac{V_{r}\;\left(t_{1}+t_{c}\right)}{\tau \;\left(1-e^{{-t}_{1}/\tau }\right)}\right]}^{2}\frac{C_{out}\;\left(1-e^{-2t_{1}/\tau }\right)}{2}$$
with the first derivative:(27)$$\frac{{dW}_{p\_th}}{{dt}_{1}}=\frac{C_{out}\;V_{r}^{2}\;\left(t_{1}+t_{c}\right)\left[\tau {\;e}^{2t_{1}/\tau }+(-t_{1}-t_{c}){\;e}^{t_{1}/\tau }-\tau \right]}{\tau^{3}\;{\left(e^{t_{1}/\tau }-1\right)}^{2}}$$

Since all parameters included in the equation are positive numbers, the derivative equals 0 if:(28)$$\left[\tau {\;e}^{2t_{1}/\tau }+(-t_{1}-t_{c}){\;e}^{t_{1}/\tau }-\tau \right]=0$$

The solution to Equation (28), which is independent of the rheobase, has been found by the numerical iterative approach, assuming τ = 1.6 or 2 ms and chronaxie increasing in 0.1 ms steps from 0.1 to 1 ms. The website https://www.derivative-calculator.net/ (David Scherfgen IT Services, Germany, accessed in 2023–2024) was chosen as a source of reliable calculation tools. The obtained results are shown in Figure 4a (open symbols): with exponentially decaying waveforms, the minimum threshold pulse energy is achieved at a duration shorter than the chronaxie. The relative difference is small, yet not negligible, and increases if the chronaxie increases.

With current implantable stimulators, furthermore, the energy released to the load per cardiac cycle does not depend on pulse energy alone but on the sum of pulse energy and the energy stored in the isolation capacitors. By defining $V_{p}$ as $V_{p\_th}$ according to Equation (25), and inserting it into Equation (21), the threshold released energy as a function of pulse duration is as follows:(29)$$W_{l\_th}(t_{1})=\frac{C_{out}}{2}{\left[\frac{V_{r}\;\left(t_{1}+t_{c}\right)}{\tau \;\left(1-e^{{-t}_{1}/\tau }\right)}\right]}^{2}\left[\left(1-e^{-2t_{1}/\tau }\right)+\frac{C_{out}{\;\left(1-e^{{-t}_{1}/\tau }\right)}^{2}}{C_{a,c}}\right]$$
(Figure 3, solid curve). This function was studied as described above, in the same conditions (τ = 1.6 or 2 ms, with increasing chronaxie), further assuming $C_{a,c}$ and $C_{out}$ = 5 and 4 µF, respectively (Figure 4a, full symbols). Taking the energy of the isolation capacitors into consideration, the pulse duration entailing minimum energy release is remarkably shorter than the chronaxie (around 90% and 80%, respectively, if the chronaxie is 0.3 or 0.6 ms). The longer the pulse, the larger the charge stored on the isolation capacitors, and correspondingly, the higher the contribution of the charge balancing to the released energy. Nevertheless, setting a pulse duration equal to the chronaxie is still good praxis in common clinical conditions, since it generally lies within the duration range where the released energy is low and mildly affected (Figure 3). In the provided example, pacing at the chronaxie, instead of the optimal duration, increases the released energy by less than 2% with chronaxie values up to 0.6 ms (Figure 4b). It must be stressed, anyhow, that the chronaxie is close to the upper limit of the advisable duration, beyond which the energy curve becomes increasingly steeper, and pulse elongation is no more cost-effective. A duration exceeding the chronaxie is generally not recommended [3,5,6,7].

## 5. Pacing with a Safety Margin

Though pacing at the threshold voltage $\left(V_{p\_th}\right)$ should be sufficient for target activation, the output of a pacing device is normally programmed to a higher amplitude to prevent frequent capture failure due to threshold changes, which can be induced by postural, physiological, or pharmacological factors. The stimulation safety margin can be defined in relative terms or by the absolute difference between pulse and threshold start amplitude. The former specification has classically been used in pulse amplitude manual programming, while the latter is often applied in algorithms of amplitude autoregulation aimed at energy saving [2].

For either the pulse energy (Equation (19)) or the total energy released to the load (Equation (21)), if all parameters but the pulse amplitude are constant, the ratio versus threshold is simply the following:(30)$$\frac{W_{p}(t_{1})}{W_{p\_th}(t_{1})}=\frac{W_{l}(t_{1})}{W_{l\_th}(t_{1})}={\left(\frac{V_{p}}{V_{p\_th}}\right)}^{2}=S_{rel}^{2}$$
where $S_{rel}=$ $V_{p}/\;V_{p\_th}$ is the relative safety margin. Therefore,(31)$$W_{l}\left(t_{1}\right)=W_{l\_th}\left(t_{1}\right)\;S_{rel}^{2}$$

Pacing with a relative safety margin increases the energy released with respect to threshold pacing by a factor equal to the square of the relative margin itself. Introducing a relative safety margin does not affect the relation between energy and pulse duration, which is simply scaled up by $S_{rel}^{2}$. As a result, the pulse duration which minimizes the energy expense remains the same as at the threshold. Furthermore, from Equation (14),(32)$$S_{rel}=\frac{V_{p}}{V_{p\_th}}=\frac{V_{m\;}t_{1}}{\tau \;\left(1-e^{{-t}_{1}/\tau }\right)}*\frac{\tau \;\left(1-e^{{-t}_{1}/\tau }\right)}{V_{m\_th\;}t_{1}}=\frac{V_{m}}{V_{m\_th}}$$
the relative margin of the pulse start amplitude reflects the ratio of the mean amplitude as well.

If an additive margin $S_{add}$ is applied to the start amplitude $\left(V_{p}=V_{p\_th}+S_{add}\right)$,(33)$$\frac{W_{p}(t_{1})}{W_{p\_th}(t_{1})}=\frac{W_{l}(t_{1})}{W_{l\_th}(t_{1})}={\left(\frac{V_{p\_th}(t_{1})+S_{add}}{V_{p\_th}(t_{1})}\right)}^{2}={\left(1+\frac{S_{add}}{V_{p\_th}(t_{1})}\right)}^{2}$$

Both the pulse and the released energy increase versus threshold pacing as a polynomial function of the safety margin. Hence,(34)$$W_{l}\left(t_{1}\right)=W_{l\_th}\left(t_{1}\right)\;{\left(1+\frac{S_{add}}{V_{p\_th}(t_{1})}\right)}^{2}$$

From the definitions of $W_{l}$ and $V_{p\_th}$ provided by Equations (21) and (25), respectively, it follows that,(35)$$W_{l}(t_{1})=\frac{C_{out}}{2}{\left[\frac{V_{r}\;\left(t_{1}+t_{c}\right)}{\tau \;\left(1-e^{{-t}_{1}/\tau }\right)}+S_{add}\right]}^{2}\left[\left(1-e^{-2t_{1}/\tau }\right)+\frac{C_{out}{\;\left(1-e^{{-t}_{1}/\tau }\right)}^{2}}{C_{a,c}}\right]$$
which corresponds to Equation (29) in the absence of a safety margin. While a relative margin increases the released energy without affecting the pulse duration entailing minimum consumption, an additive margin progressively shortens it and shrinks the advisable duration range (Figure 5). The prediction for an implant featuring $I_{r}=1\;mA$, $t_{c}=0.4\;ms$, $C_{out}=4\;µF$, $C_{a,c}=5\;µF$, R=500 or 400 Ohm (that is, respectively, $V_{r}=0.5\;or\;0.4\;V;\;\tau =2\;or\;1.6\;ms$), and $S_{add}$ ranging from 0 to 0.6 V, is provided in Figure 6. The optimal duration is shorter than the chronaxie, and the difference increases with increasing additive margin (Figure 6a). The energy increment required to pace at the chronaxie, instead of the optimal value, increases with the additive margin as well (Figure 6b). As an example, with a 0.5 V additive margin, the optimal duration is reduced to less than 0.2 ms, and the energy released with a duration equal to the chronaxie exceeds the minimum by about 15%, depending on the time-constant.

If an implantable stimulator is designed to pace with a constant increment with respect to the threshold amplitude, independent of pulse duration, maximum energy saving is achieved at a duration shorter than the chronaxie. This prediction has been confirmed by a recent clinical study on left bundle branch permanent pacing, where an additive safety margin of 1 V was applied. A pulse duration of 0.2 ms entailed the minimum battery consumption in front of a 0.38 ms chronaxie [12]. Some caution is advisable, however, since the threshold amplitude and duration are inversely related, and a fixed additive margin necessarily implies a decreasing relative margin as the pacing pulse gets shorter and shorter.

## 6. Pulse Duration and Pacing Effectiveness

Lapicque’s law states that the mean amplitude of the threshold pulse $\left(V_{m\_th}\right)$ decreases as a function of pulse duration, the relation being steeper for short pulses and progressively flatter as the duration increases. The safety margin of a pulse with stable amplitude is thus always enhanced by the elongation of the application time. However, the mean voltage $\left(V_{m}\right)$ of a pulse featuring exponential decay is not constant: it also decreases with increasing duration, as described by Equation (11). Prolonging a pulse with definite start amplitude $\left(V_{p}\right)$ increases the safety margin as long as the reduction in mean amplitude is smaller than the reduction in threshold (Figure 7a). If the stimulation safety is expressed in relative terms,(36)$$\frac{V_{m\;}}{V_{m\_th}}=\frac{V_{p\;}}{V_{r\;}}*\frac{\tau \;\left(1-e^{{-t}_{1}/\tau }\right)}{\left(t_{1}+t_{c}\right)}$$
where $V_{m}$ and $V_{m\_th}$ are defined according to Equations (11) and (2), respectively. Provided that $V_{p}$, $V_{r}$, τ, and $t_{c}$ are constant, this ratio is a function of pulse duration, which equals 1 at threshold and reaches the maximum when the first derivative equals 0, with negative second derivative:(37)$$\frac{{d(V}_{m}/V_{m\_th})}{{dt}_{1}}=-\frac{V_{p\;}}{V_{r\;}}*\frac{e^{-t_{1}/\tau }\left({\tau \;e}^{t_{1}/\tau }-t_{1}-\tau -t_{c}\right)\;}{{\left(t_{1}+t_{c}\right)}^{2}}=0$$

Since $V_{p}>0$ and $t_{1}$ is a finite number:(38)$$\tau \;e^{t_{1}/\tau }-t_{1}-\tau -t_{c}=0$$

On the other hand, if additive safety is concerned:(39)$$V_{m}-V_{m\_th}=\frac{\tau \;V_{p\;}\left(1-e^{{-t}_{1}/\tau }\right)}{t_{1}}-V_{r}\left(1+\frac{t_{c}}{t_{1}}\right)=V_{r}\left[\frac{V_{p\;}}{V_{r}}*\frac{\;\tau \;\left(1-e^{{-t}_{1}/\tau }\right)\;}{\;t_{1}}-\frac{t_{c}}{t_{1}}-1\right]$$
which equals 0 at threshold. In such an instance, since $V_{r}$ is positive:(40)$$\frac{V_{p\;}}{V_{r}}*\frac{\;\tau \;\left(1-e^{{-t}_{1}/\tau }\right)}{\;t_{1}}-\frac{t_{c}}{t_{1}}-1=0$$

Maximum additive safety is achieved when the first derivative of Equation (39) is 0, and its second derivative is negative:(41)$$\frac{{d(V}_{m}-V_{m\_th})}{{dt}_{1}}=\frac{V_{r\;}e^{-t_{1}/\tau }\left[e^{t_{1}/\tau }\left(t_{c}-\tau \frac{V_{p\;}}{V_{r\;}}\right)+t_{1}\frac{V_{p\;}}{V_{r\;}}+\tau \frac{V_{p\;}}{V_{r\;}}\right]\;}{t_{1}^{2}}=0$$

Again, with positive $V_{r}$ and $0<t_{1}<\infty$,(42)$$\frac{V_{p\;}}{V_{r\;}}\left[\tau \;\left(1-e^{t_{1}/\tau }\right)+t_{1}\right]+t_{c\;}e^{t_{1}/\tau }=0$$

The $t_{1}$ values for which the above statements hold true, worked out by iterative numerical solution as a function of the ratio between pulse start amplitude and rheobase $\left(V_{p}/V_{r}\right)$, are reported in Figure 7b. The curves represent the pulse application time required to stimulate at the threshold (no safety margin) or with maximum relative or additive safety by assuming chronaxie and time-constant equal to 0.4 and 1.6 ms, respectively. Any amplitude–duration combination lying below the threshold curve has no physiological effect, as the critical membrane depolarization is not achieved and capture is missing. Any combination lying above the maximum safety curves does not increase the safety margin, therefore, exceeding the longest useful duration is pure energy waste. It is noteworthy that the longest useful duration for relative safety is independent of both rheobase and start pulse amplitude (Equation (38)). This implies that a threshold rise induced by rheobase increase can be counteracted by pulse elongation up to the maximum useful duration only: any further increase is ineffective if the chronaxie remains constant. In contrast, the pulse duration at which the highest additive safety is achieved increases with increasing rheobase or decreasing start amplitude up to the duration of maximum relative safety (Equation (42)). If the $V_{p}/V_{r}$ ratio increases, the longest duration that can improve the additive safety decreases, becoming shorter than 0.7 ms with $V_{p}/V_{r}>3.5$. The ordinate where the three curves overlap indicates the effective duration of a pulse featuring the lowest threshold start amplitude (minimum $V_{p}/V_{r}$ allowing capture without safety margin): by no way a smaller pulse, whatever long, can produce successful stimulation. The lowest threshold start amplitude and the corresponding mean pulse amplitude are both remarkably higher than the rheobase $\left(V_{p}/V_{r}>1\right)$, which cannot be approached in practice by a declining pulse with a time-constant of a few milliseconds. The rheobase corresponds instead to the instantaneous amplitude of the lowest threshold pulse at the unique duration that allows effective stimulation, according to the time-course of pulse decay described by Equation (8) [4].

As a practical example, Figure 7a shows Lapicque’s curve and pulse mean amplitude as a function of duration by assuming 0.4 V rheobase, 0.4 ms chronaxie, and 1.6 ms time-constant. If the pulse start amplitude is 1 V ${(V}_{p}/V_{r}=2.5)$, the stimulation is ineffective with any duration shorter than 0.32 ms. From this point, the pulse mean amplitude exceeds the threshold mean amplitude, and capture is achieved with an increasing safety margin. The maximum difference between $V_{m}$ and $V_{m\_th}$ is reached at 0.85 ms duration, when the pulse mean amplitude is 0.78 V versus a threshold of 0.59 V. The relative safety is maximum at 1.01 ms, where $V_{m}/V_{m\_th}$ = 1.32 (0.74 V/0.56 V). Longer pulses entail a reduced difference or ratio, but the actual safety margin remains constant at the maximum level previously gained, as the effect produced at a shorter duration is not canceled if the pulse continues. A pulse featuring a start amplitude of 1.25 V ${(V}_{p}/V_{r}=3.12)$ crosses the threshold curve at 0.21 ms, reaches maximum additional safety at 0.75 ms duration (1 − 0.62 = 0.38 V), and maximum relative safety (0.93 V/0.56 V = 1.66) at 1.01 ms again, since the longest duration that can increase the relative safety does not depend on the pulse amplitude. The lowest threshold start amplitude is 0.75 V ${(V}_{p}/V_{r}=1.875)$: such a pulse touches the threshold curve at only one point, corresponding to 1.01 ms duration and 0.56 V mean amplitude, versus the rheobase potential of 0.4 V. Further duration increase cannot influence the pacing outcome, as $V_{m}-V_{m\_th}$ becomes increasingly negative and $V_{m}/V_{m\_th}$ increasingly <1. Any pulse with a start amplitude below 0.75 V cannot reach the threshold, regardless of its duration.

The relation between the threshold start amplitude $\left(V_{p\_th}\right)$ and pulse duration is described by Equation (25). This function decreases more slowly than Lapicque’s curve with increasing pulse duration, until the reduction stops even if the rheobase is far away. Beyond this point, the theoretical threshold start amplitude features a non-physiological increase, while the real curve remains constant at the minimum value since capture is achieved anyhow at the shortest threshold duration, and the additional stimulation time is devoid of physiological relevance (Figure 8a). The threshold pulse charge, derived from the real threshold start amplitude according to Equation (13), is a linear function of the duration (as stated by Lapicque’s law) as long as the threshold start amplitude decreases. When it becomes constant, the corresponding threshold charge deviates from the linear regression and must not be considered in the assessment of excitability, as the pulse tail is ineffective (Figure 8b).

The pulse duration at which capture is achieved with minimum threshold start amplitude (corresponding to the longest useful duration, as shown in Figure 7) must satisfy the equation:(43)$$\frac{{d(V}_{p\_th}/V_{r})\;}{{dt}_{1}}=\frac{e^{t_{1}/\tau }\;\left(\tau \;e^{t_{1}/\tau }-t_{1}-\tau -t_{c}\right)}{{\left[\tau \;\left(e^{t_{1}/\tau }-1\right)\right]}^{2}}=0$$
with positive second derivative. The numerical solution, univocally defined by chronaxie and time-constant as in Equation (38), is reported in Figure 9a. The effective duration limit becomes shorter with shortening chronaxie and time-constant, ranging from 0.74 to 1.51 ms and from 0.83 to 1.72 ms, respectively, if the time-constant is set at 1.6 or 2 ms and the chronaxie ranges from 0.2 to 1 ms. The minimum value of the ratio $V_{p\_th}/V_{r}$ was then determined by inserting the longest useful duration, along with the related $t_{c}$ and τ, into Equation (25). The result increases with increasing chronaxie and decreasing time-constant, being remarkably higher than 1 throughout the physiological chronaxie range. The pulse mean amplitude at the longest useful duration largely exceeds the rheobase as well, which can never be reached, or even approached, with an implantable stimulator relying on standard current technology (Figure 9b).

The clinician should be aware that the threshold start amplitude of decaying pulses is less sensitive to pulse duration than Lapicque’s law expectation (which applies to the mean amplitude of the threshold pulse) and remains stable beyond a duration limit depending on the chronaxie and the pulse time-constant. A threshold start amplitude that does not decrease in spite of increasing duration (and the related mean amplitude) must not be mistaken for the rheobase, which is substantially lower (Figure 7, Figure 8 and Figure 9). Unless the chronaxie is abnormally long, the stimulation safety is poorly improved by pulse elongation beyond 0.8 ms, which is unlikely cost-effective. Further pulse extension beyond the longest useful duration produces no physiological effect, substantially increasing the pacing consumption.

## 7. Effects of Load Resistance on the Pacing Energy

According to Lapicque, the minimal requirement to achieve effective stimulation is the application of the threshold mean current for the selected pulse duration, that is, the release of the appropriate charge to the excitable tissue in due time. The pulse voltage does not play a direct role in the stimulation process, being just the measure of the electric potential required to generate the current [5]. The pacing energy, in contrast, is strongly dependent on the pulse voltage (Equation (21)). Therefore, producing the needed current $\left(I\right)$ at a lower voltage $\left(V\right)$ would be a good strategy to reduce the consumption and increase the device longevity without reducing the pacing efficacy. The task can be pursued by decreasing the resistance to current flow offered by the pacing lead, which is a relevant part of the total load resistance (R) since V=I∗R by Ohm’s law.

By normalizing the load resistance to a reference value so that $R/R_{ref}=k$, and expressing the start potential as a function of the mean current (Equation (14)), Equation (21) can be rewritten as follows:(44)$$W_{l}\left(k,t_{1}\right)=\frac{{\left(I_{m}\;t_{1}\right)}^{2}}{2{\;C}_{out}\;{\left(1-e^{{-t}_{1}/{k\tau }_{ref}}\right)}^{2}}\;\left[\left(1-e^{-2t_{1}/{k\tau }_{ref}}\right)+\frac{C_{out}{\;\left(1-e^{{-t}_{1}/{k\tau }_{ref}}\right)}^{2}}{C_{a,c}}\right]\;$$
where $\tau_{ref}=C_{out}{\;R}_{ref}$. If mean current and duration are kept constant (as well as $C_{out}$ and $C_{a,c}$, which cannot be modified by reprogramming):(45)$$\frac{W_{l}({k,\;t}_{1})}{W_{l\_ref}(t_{1})}={\left[\frac{\left(1-e^{{-t}_{1}/\tau_{ref}}\right)}{\left(1-e^{{-t}_{1}/{k\tau }_{ref}}\right)}\right]}^{2}\left[\frac{C_{a,c}\;\left(1-e^{-2t_{1}/{k\tau }_{ref}}\right)+C_{out}{\;\left(1-e^{{-t}_{1}/{k\tau }_{ref}}\right)}^{2}}{C_{a,c}\;\left(1-e^{-2t_{1}/\tau_{ref}}\right)+C_{out}{\;\left(1-e^{{-t}_{1}/\tau_{ref}}\right)}^{2}}\right]\;$$
where $W_{l\_ref}$ is the energy released when the load resistance is $R_{ref}$. The energy ratio is an almost linear function of k, with slope slightly decreasing with increasing pulse duration (Figure 10). If the duration tends to 0, or in the absence of pulse decay (squared pulse waveform), any fractional change in load resistance entails an equal fractional modification in the released energy. In such an instance, from Equation (22),(46)$$\frac{W_{p{}^{\circ}}\left({k,\;t}_{1}\right)}{W_{p{}^{\circ}\_ref}\left(t_{1}\right)}=\frac{k\;R_{ref}\;I_{p{}^{\circ}}^{2}\;t_{1}}{R_{ref}\;I_{p{}^{\circ}}^{2}\;t_{1}}=k$$

With implantable pacemaker pulses, the load effect is only mildly lower in the range of physiological resistance. As an example, if the resistance is lowered from 500 to 400 Ohm (20% reduction) with a pulse duration of 0.4 ms, the released energy decreases by 18%.

Reducing the load is thus an effective way to save energy, even after the pacing lead is placed in the final position. As the pacing impedance is generally smaller in unipolar than bipolar mode, the former should be preferred unless unipolar stimulation is contraindicated due to side effects, such as pectoralis muscle twitch. This is rarely a true concern in low-energy pacing; nevertheless, the bipolar modality is often the first choice in the clinical praxis. The clinician should consider that bipolar stimulation can easily entail a substantial increase in pacing consumption (10 to 20%), essentially due to the increase in total load impedance resulting from the inclusion of the ring electrode and the return coil in the current pathway [13]. The pacing polarity is not expected to produce relevant effects on the conduction process at the interface between the cathode and the surrounding tissue, where the threshold current should be virtually unaffected [1]. In contrast, the current distribution at a distance from the source might be modified by a polarity change, which could thus influence the stimulation performance if the pacing electrode is relatively far from the target (e.g., myocardial stimulation from a cardiac vein or through a wide layer of fibrotic tissue, non-selective His bundle pacing). This possibility should be checked and taken into consideration when the pacing configuration is set.

Though the lead resistance should be kept as low as possible to improve the stimulation efficiency, an increase in overall pacing impedance could be beneficial if produced by a reduction in the electrode surface. The depolarizing effect of a stimulus depends on the intensity of the electric field produced at any point in the conducting medium, which is the product of local current density and specific resistivity [2,5,14]. A smaller tip electrode increases the current density (and the associated electric field) in the surroundings of the cathode, allowing effective pacing by the release of lower total current. This results in energy saving, provided that the associated impedance rise does not entail an excessive increase in threshold voltage [15,16]. Actually, a geometric electrode area of about 1 mm^2^ seems the lowest limit for successful clinical use [17]. Further surface reduction might, in addition, worsen the pulse conduction by decreasing the Helmholtz capacitance, which is the main factor allowing the current to flow from the pacing lead into the stimulated tissue [2,13].

## 8. Role of the Electrode Capacitance

The interface between the electrode and the surrounding conducting volume, which is essentially a water solution containing charged ions, is generally represented as a capacitance and a resistance connected in parallel. However, if the electric potential is below a few Volts, as expected in cardiac pacing, the resistive component is virtually negligible, and the current conduction is almost totally dependent on the electrode capacitance, generally referred to as Helmholtz capacitance $\left(C_{H}\right)$. Indeed, the stimulating current can flow across the tissue because the ionic charge built at the liquid conductor interface is balanced by an equal and opposite electronic charge stored on the electrode surface $\left(Q_{H}\right)$. Positive and negative charges are separated by a double layer of water molecules, acting as a dielectric to provide the electric capacitance $C_{H}=Q_{H}/V_{H}$, where $V_{H}$ is the potential resulting from electrode polarization. Since $V_{H}$ represents a counter-electromotive force that reduces the pulse current, the higher the electrode capacitance, the better the current conduction, even with long-duration pulses [2,13].

A great effort has been devoted to the development of pacing electrodes featuring high capacitance in spite of the small geometric area. This has been achieved by increasing the effective solid–liquid interface, thanks to the use of microporous or fractal materials [17,18,19]. In general, electrode polarization during pulse conduction should no longer be an issue in cardiac pacing, hence the load can be practically considered purely resistive, as in the whole above description [13]. However, the implications of a limited (however big) electrode capacitance cannot be ignored in the case of a large pulse charge, which is required when the target is poorly excitable or not in close contact with the current source due to anatomical constraints or fibrosis in the implant region.

While the anode capacitance is supposed to be large enough to be ignored, thanks to the wide electrode surface, the smaller tip electrode capacitance in series with the stimulator output stage might reduce the overall equivalent capacitance $\left(C_{tot}\right)$:(47)$$C_{tot}=\frac{1}{\frac{1}{C_{out}}+\frac{1}{C_{H}}}=\frac{1}{\frac{1}{C_{res}}+\frac{1}{C_{c}}+\frac{1}{C_{a}}+\frac{1}{C_{H}}}$$

When this is the case, the time course of the pulse current and related voltage applied to the load is described as in Equations (8)–(10) by replacing $C_{out}$ with $C_{tot}$. The reduced overall capacitance results in a shorter time-constant and correspondingly reduced pulse charge, the difference increasing with increasing pulse duration. Relying on the purely resistive model, if the electrode capacitance is too small, would raise the slope of the relation between threshold charge and duration, leading to rheobase overestimation and chronaxie underestimation. Unfortunately, the electrode capacitance is not measured by conventional stimulators, so the actual $C_{H}$ value in situ is usually unknown. The average Helmholtz capacitance in standard conditions, as declared by the manufacturer of the pacing lead, is the only piece of information available to establish whether the electrode polarization influence on pulse conduction is negligible.

## 9. A Practical Approach to Pulse Optimization in Cardiac Pacing

Though Lapicque’s law applies to any pulse waveform, the most popular concepts derived from it should properly be restricted to the application of squared pulses, which is just an ideal assumption in permanent clinical pacing. As a matter of fact, the stimuli actually delivered by implantable pacemakers feature an amplitude decay with exponential time-course, depending on the device output capacitance and the load resistance. Furthermore, the pulse is delivered through a couple of isolation capacitors, which are discharged when the pulse is over, thus contributing to the total energy released to the load. By considering this in theoretical prediction, the pulse duration allowing capture with minimum energy expense is shorter than the chronaxie. Setting the pacing pulse at the chronaxie time induces a minor energy increase with threshold amplitude. In contrast, a larger difference is expected in the presence of an additive safety margin [12]. Extending the duration beyond the chronaxie substantially increases the pacing consumption, with a decreasing effect on the safety margin, which does not improve at all if the longest useful duration is overshot. In such conditions, the threshold start amplitude vs. duration curve is constant at the minimum as well. At the longest useful duration, the threshold start amplitude and the corresponding mean amplitude are both higher than the rheobase (Figure 9). The longest useful duration ranges from 0.7 to 1.3 ms in most cases and cannot be extended unless the chronaxie increases. Pacing with longer pulses is often useless and always expensive: it should thus be limited to implants where the pacing electrode is far from the target, and a long chronaxie is expected or known.

Rheobase and chronaxie can be determined by assessing the start amplitude of the threshold pulse as a function of pulse duration, working out the corresponding pulse charge according to Equation (13), and performing a linear regression analysis of threshold charge on duration. The procedure can be carried out by any commercial spreadsheet but requires knowledge of the stimulator output capacitance. Should official information not be available, the output capacitance can be derived, with acceptable approximation, from the device pulse waveform recorded in nominal pacing configuration with a standard load. From the relation between instantaneous pulse amplitude and application time, described by Equation (8), it follows that:(48)$$\ln V(t)=\ln V_{p}-t/\tau =\ln V_{p}-\frac{t}{R\;C_{out}}$$(49)$$C_{out}=\frac{t}{R\;\left[\ln V_{p}-\ln V(t)\right]}$$

It can be convenient to choose ${t=t}_{1}$ (the whole pulse duration) and measuring the output voltage at pulse leading $\left(V_{p}\right)$ and trailing edge $\left(V_{1}\right)$ to calculate the difference of their natural logarithms. As both the standard load resistance and pulse duration are known, the standard output capacitance can readily be derived for any pacemaker model. If time and impedance are expressed in ms and Ohm, respectively, the result must be multiplied by 10^3^ to obtain the capacitance in µF. On the other hand, if the capacitance is scaled in µF, the time-constant $\tau ={R\;C}_{out}$ must be divided by 10^3^ to be converted into ms, that is, the same unit of measure usually applied to pulse duration and chronaxie. With capacitance in µF, all time measures ($t_{1}$, τ, and $t_{c}$) in ms, and pulse voltage in V, the pulse charge is obtained in µC. The slope of the threshold charge–duration line (µC/ms), corresponding to the rheobase current, is thus expressed in mA.

As stated above, the pulse duration should never exceed the chronaxie to reduce the pacing energy. The optimal duration might rather be lower and increasingly shorter if a constant safety margin is added to the threshold amplitude. The latter issue should be taken into special consideration, as many systems of automatic amplitude regulation work this way [2]. In His bundle pacing, a short pulse duration might favor selective capture, as the chronaxie of the surrounding myocardium is generally longer [20,21]. Stimulation with very short pulses (0.03 ms) allowed selective activation of the conduction system during the threshold test in a substantial proportion of patients treated with left bundle branch pacing [12]. Conversely, reducing the pulse amplitude thanks to a duration increase might prevent phrenic nerve stimulation in CRT [22,23,24], producing a clinical benefit that justifies larger energy expenses.

Unipolar pacing should be considered as a solution to curtail consumption, whenever it reduces the impedance by 10% or more in the absence of side effects. The shorter the pulse, the higher the benefit, as shown in Figure 10. Both the unipolar and bipolar pacing thresholds should be assessed with the same pulse duration. If the impedance is different, the threshold mean voltage should correspondingly be different (generally lower in unipolar mode), while the threshold mean current and corresponding pulse charge are supposed to be unaffected, provided that the stimulating electrode is close to the target [13,25,26]. Pulse mean current and charge are not generally measured by implantable pacemakers featuring voltage-regulated output but can quickly be calculated, as for Equations (12) and (13), in the absence of significant electrode polarization.

Pulse optimization is increasingly important in the case of large pacing consumption, which is not rare in endocardial pacing and much more frequent in epicardial or His bundle stimulation [27,28]. Individual tuning of pulse duration and reduction in pacing impedance can substantially contribute to energy saving in the presence of a high threshold. The impact on device longevity depends on the ratio of pacing versus total current drained from the battery, which can be affected, in turn, by the stimulator design, the pacing lead properties, the stimulation target (myocardium or conduction system), and the position of the pacing electrode with respect to the target. The practical value of the theoretical evidence should be tested in clinical studies. Preliminary experience confirms that the efficiency of His bundle pacing is substantially improved by appropriate pulse setting, based on avoidance of unnecessary long pulses, selection of the most convenient pacing polarity, and safety margin reduction thanks to reliable capture surveillance [20,29,30,31].

## 10. Limitations

This paper intentionally deals with theoretical predictions only. Assessing their relevance in clinical practice is a matter for future research. The study evaluates the combined effects of safety margin, chronaxie, pulse duration, and time-constant, in a range of values that apply to myocardium and conduction system pacing in normal conditions. The simulations do not include the analysis of extreme changes in pacing impedance due to lead failure.

## 11. Conclusions

As dual-chamber, three-chamber, and multifocal pacing are becoming more and more common in clinical practice, stimulation efficiency plays an increasing role in the restraint of device consumption. In addition to progress in the implantable stimulator and pacing lead design, accurate lead positioning on implantation and optimal pulse programming are required to pursue the task. In particular, the pulse duration can substantially influence the pacing energy required for effective stimulation. Pulse waveform and charge balancing should be taken into consideration in a rational duration setting. The decaying pulses actually released by implantable stimulators feature a maximum useful duration. Their threshold start amplitude can never approach the rheobase, unlike what is expected with squared pulses, which can be produced by external stimulators only. Pacing with long pulses should be restricted to special cases characterized by unusually long chronaxie. A pulse duration shorter than the chronaxie is advisable for energy saving, especially if an additive safety margin is applied. Switching from bipolar to unipolar pacing can also be helpful in reducing the released energy while keeping the stimulating current constant.

Careful management of pulse amplitude and duration, combined with the preference for unipolar stimulation whenever possible, can remarkably decrease the current drained from the battery with no compromise on the quality of the pacing therapy. This would prolong the service life of the stimulator and correspondingly reduce the replacement frequency, with substantial benefits for patients and the global healthcare system.

## Figures and Tables

**Figure 1 bioengineering-12-00194-f001:**
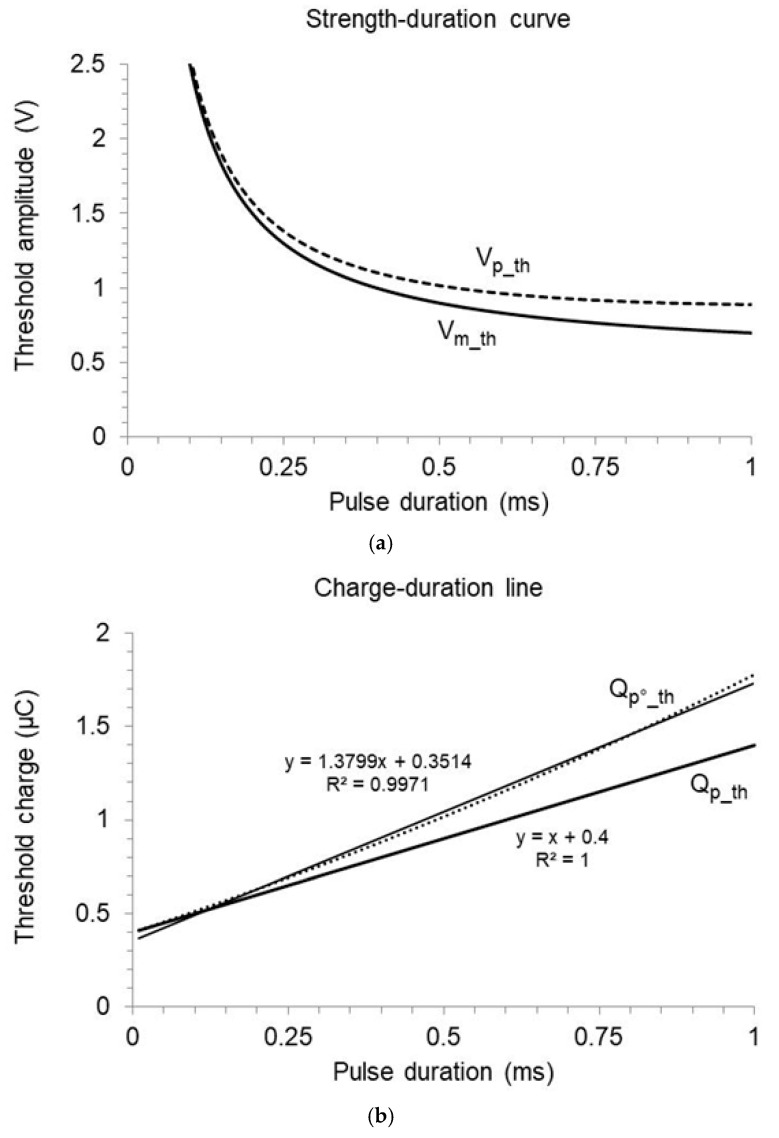
Different outcomes in excitability assessment considering or ignoring the exponential pulse decay. The simulation assumes $I_{r}=1\;mA$, $t_{c}=0.4\;ms$, R=500 Ohm, $C_{out}=4\;µF$. (**a**) Mean voltage of the threshold pulse as a function of pulse duration (Lapicque’s law, solid curve) and corresponding start voltage, derived from the exponential decay model (dashed curve). (**b**) Linear regression of threshold charge on duration based on the mean or the start pulse voltage (thick and light lines, respectively).

**Figure 2 bioengineering-12-00194-f002:**
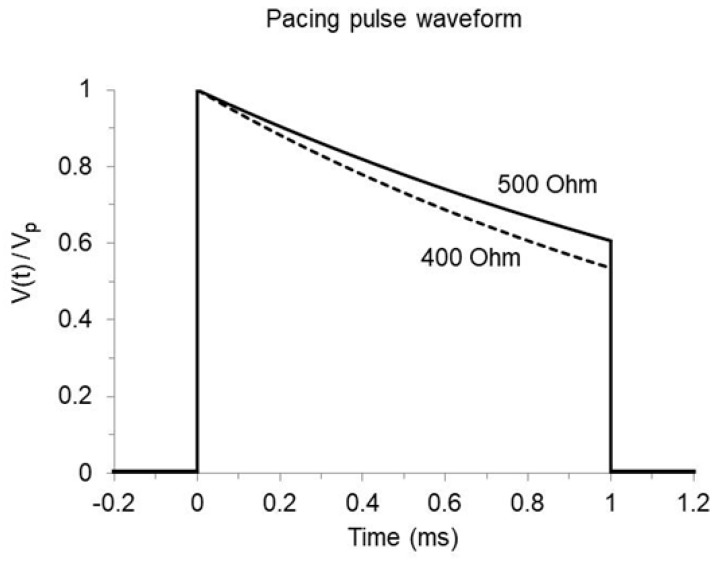
Time-course of pulse voltage $\left(V(t)\right)$ normalized to the start value $\left(V_{p}\right)$ in an implantable pacemaker featuring 4 µF output capacitance and 500 or 400 Ohm load resistance (solid and dashed curves, respectively). The voltage declines exponentially during the stimulus at a rate that increases if the product of output capacitance and load resistance decreases. Though the amplitude ratio is positive, it must be pointed out that the pacing stimulus is actually a cathodal pulse (the tip electrode is negative with respect to either the ring electrode or the pacemaker case).

**Figure 3 bioengineering-12-00194-f003:**
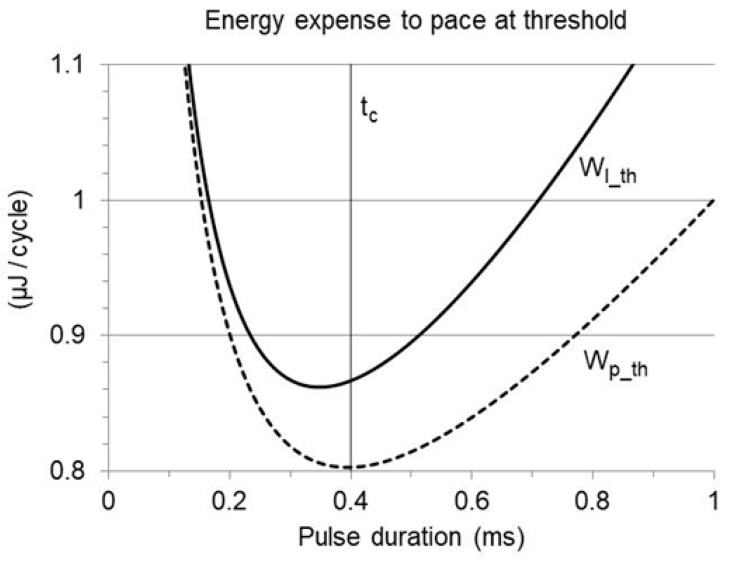
Energy expense per cardiac cycle to pace at threshold in an implant with $I_{r}$ = 1 mA, $t_{c}$ = 0.4 ms, $C_{out}$ = 4 µF, $C_{a,c}$ = 5 µF, R = 500 Ohm, considering the pulse energy alone ($W_{p\_th}$: dashed curve) or the total energy released to the load by spike emission and charge balancing ($W_{l\_th}$: solid curve). The latter is minimized at a pulse duration shorter than the chronaxie ($t_{c}$: vertical line).

**Figure 4 bioengineering-12-00194-f004:**
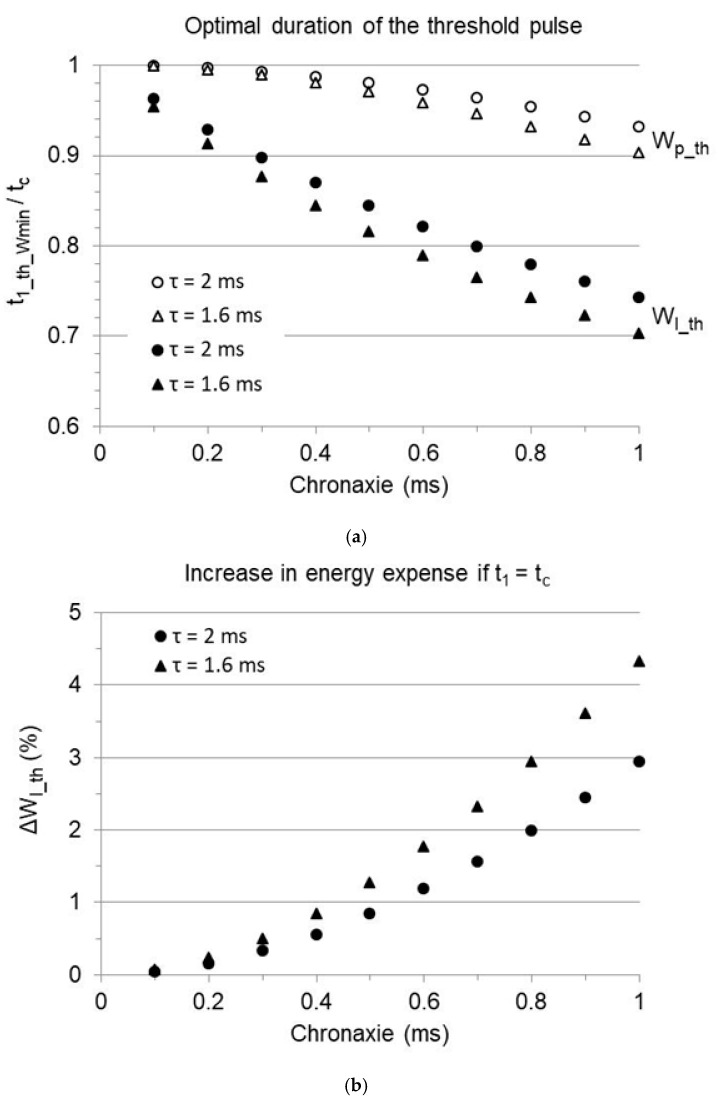
The same simulation as in Figure 3, assuming a load resistance of either 500 Ohm ($V_{r}=0.5\;V;\;\tau$ = 2 ms) or 400 Ohm ($V_{r}=0.4\;V;\;\tau$ = 1.6 ms) and variable chronaxie. (**a**) The pulse duration implying minimum energy expense to pace at threshold, normalized to the chronaxie time, is plotted versus the chronaxie, considering the pulse energy alone ($W_{p\_th}$: open symbols) or the total energy released to the load per cardiac cycle ($W_{l\_th}$: full symbols). Both are minimized at a pulse duration shorter than the chronaxie, though the difference is much more prominent for the released energy. (**b**) Percent increase in energy released to the load to pace at threshold, if the pulse duration is set at the chronaxie instead of the optimal value.

**Figure 5 bioengineering-12-00194-f005:**
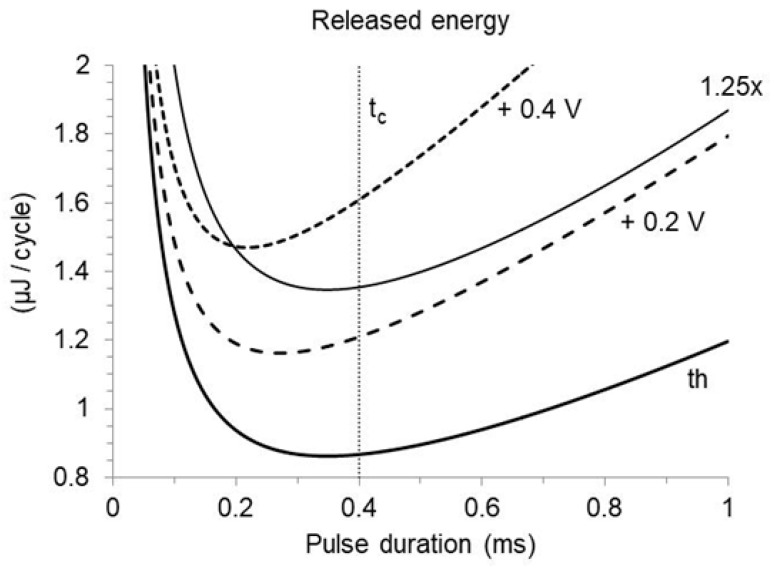
Same simulation as in Figure 3, comparing the energy released to the load per cardiac cycle at threshold (th) and in the presence of a relative (1.25×) or additive safety margin (+0.2 V; +0.4 V). A relative margin increases the energy without affecting its relation with the pulse duration. In contrast, an additive margin progressively shortens the duration that entails minimum energy consumption, and shrinks the advisable duration range. The vertical dotted line indicates the chronaxie time ($t_{c}$).

**Figure 6 bioengineering-12-00194-f006:**
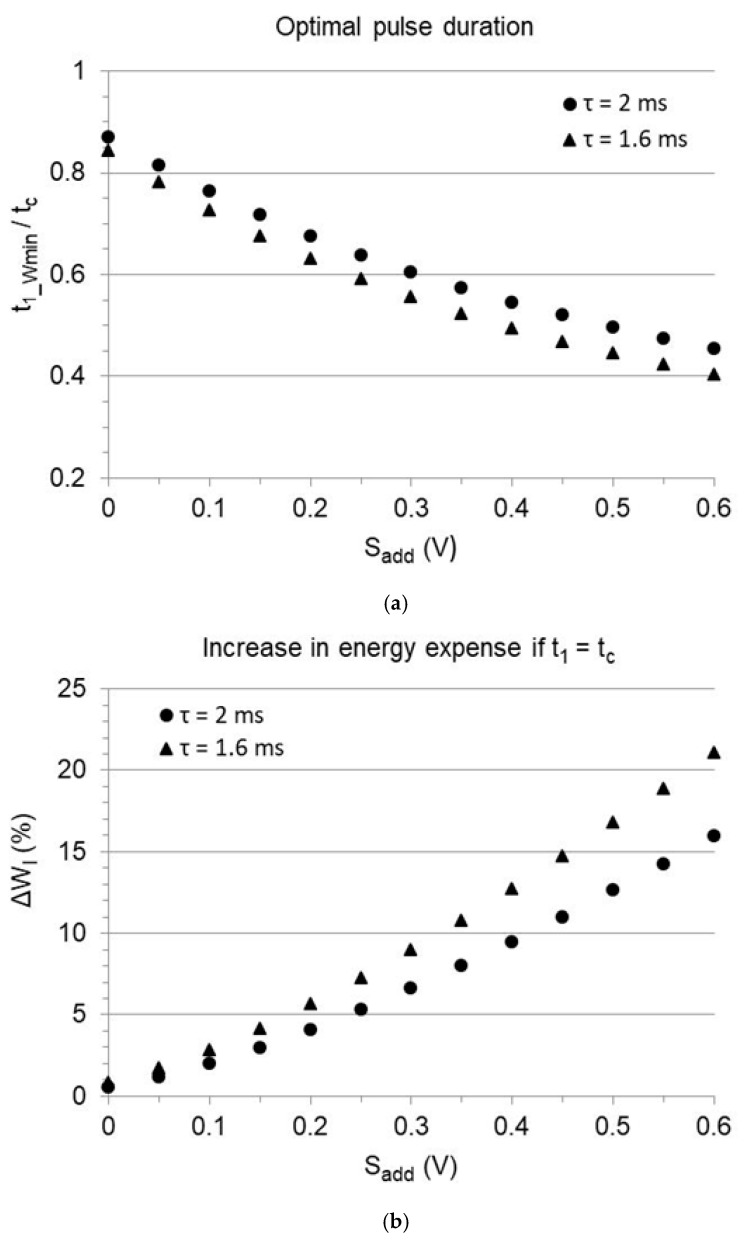
Effect of an additive safety margin ($S_{add}$) on the energy released to the load. Same simulation as in Figure 4, with 0.4 ms chronaxie ($t_{c}$). Threshold pacing is represented as $S_{add}$ = 0. (**a**) Pulse duration minimizing the released energy ($t_{1\_Wmin}$), normalized to the chronaxie and plotted as a function of the additive safety margin. (**b**) Difference between the energy released to the load at the chronaxie and at the optimal duration reported in panel (**a**), expressed as percent of the minimum energy.

**Figure 7 bioengineering-12-00194-f007:**
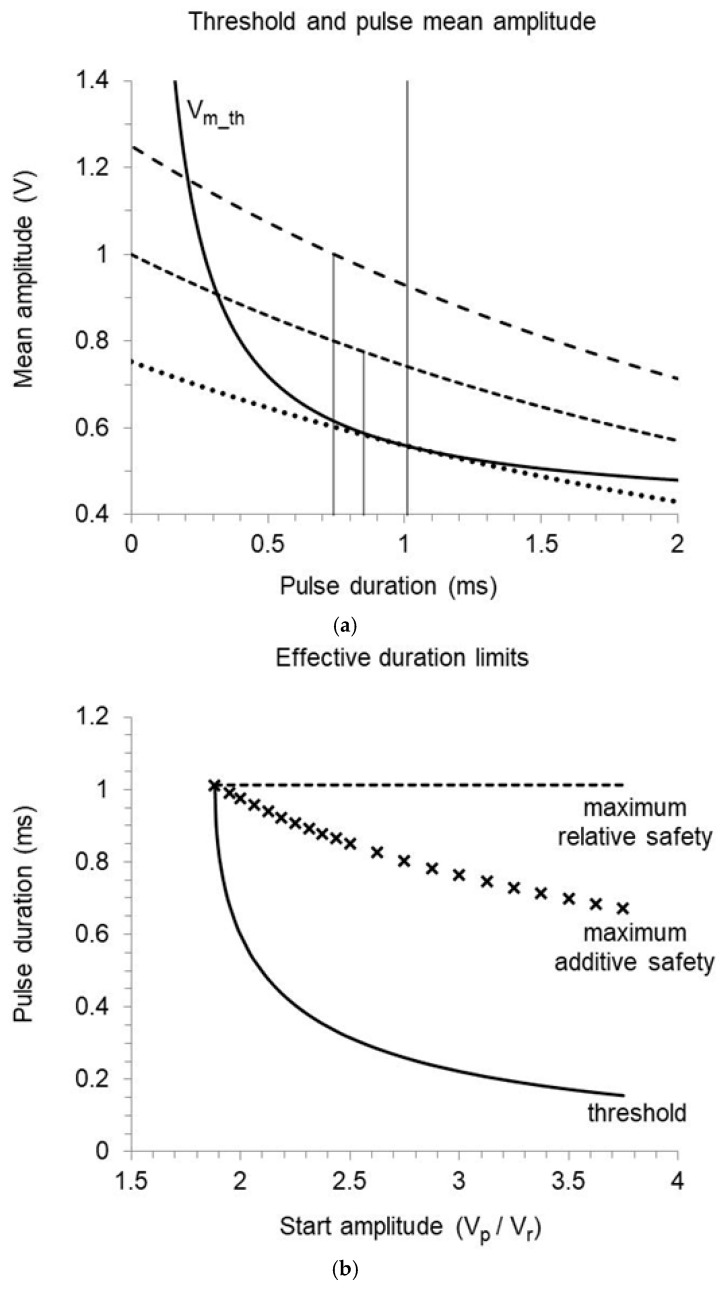
Predictions referring to an implant with 0.4 V rheobase, 0.4 ms chronaxie, and 1.6 ms time-constant. (**a**) Threshold mean voltage as a function of pulse duration according to Lapicque ($V_{m\_th}$: solid curve) compared with the mean voltage of decaying pulses with a start amplitude of 1.25 V (wide dashed curve), 1 V (narrow dashed curve), or 0.75 V (minimum threshold start amplitude, dotted curve). The vertical segments indicate the duration of maximum additive margin with 1.25 V or 1 V start amplitude. The highest vertical line indicates the longest useful duration, entailing maximum relative margin with any start amplitude. (**b**) Pulse duration required to reach the threshold (solid curve), or to pace with maximum relative or additive safety margin (dashed line and crosses, respectively), as a function of the ratio between start and rheobase voltage.

**Figure 8 bioengineering-12-00194-f008:**
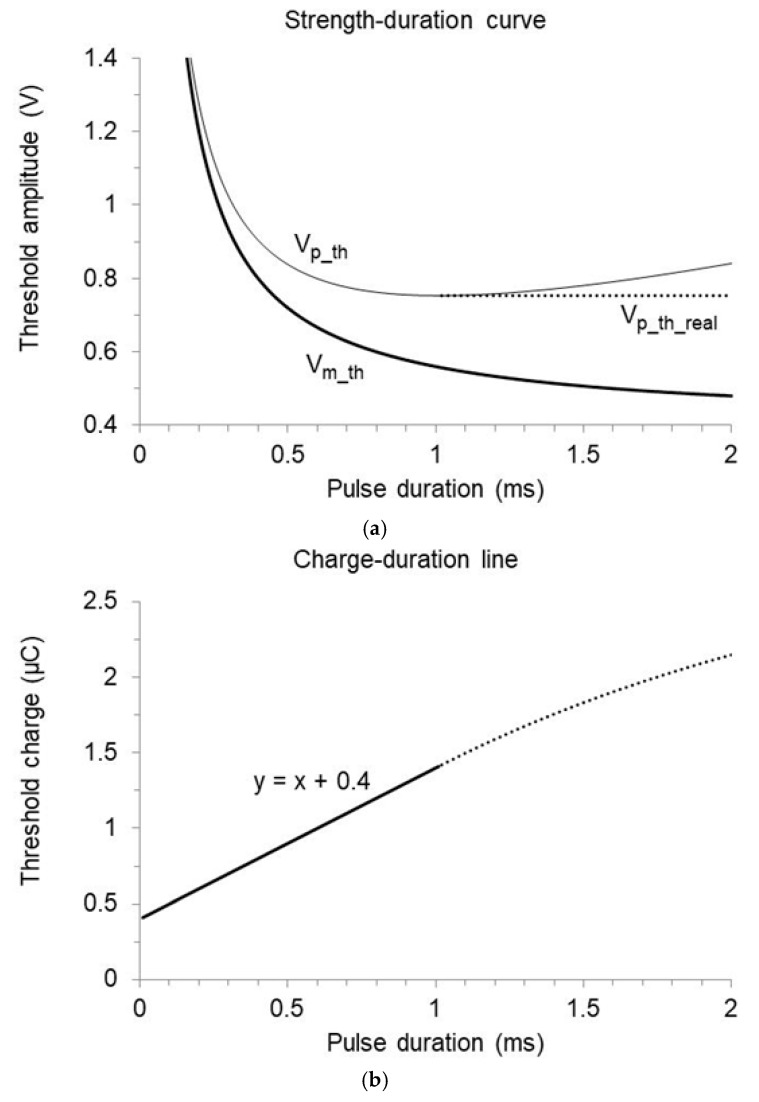
Pulse decay implications on the threshold strength–duration curve in an implant featuring $I_{r}=1\;mA$, $t_{c}=0.4\;ms$, R=400 Ohm, and $C_{out}=4\;µF$. (**a**) The threshold mean voltage decreases as a function of pulse duration (thick solid curve), tending to the rheobase $\left(V_{r}=0.4\;V\right)$. The corresponding start voltage (light solid curve) decreases with a lower slope and stops declining at a value much higher than the rheobase (0.75 V with 1.01 ms duration). Beyond this point, the actual threshold start amplitude is constant (dotted line). (**b**) The experimental relation between threshold charge and duration is linear and consistent with the assumed rheobase and chronaxie only in the duration range where the threshold start amplitude decreases (solid line).

**Figure 9 bioengineering-12-00194-f009:**
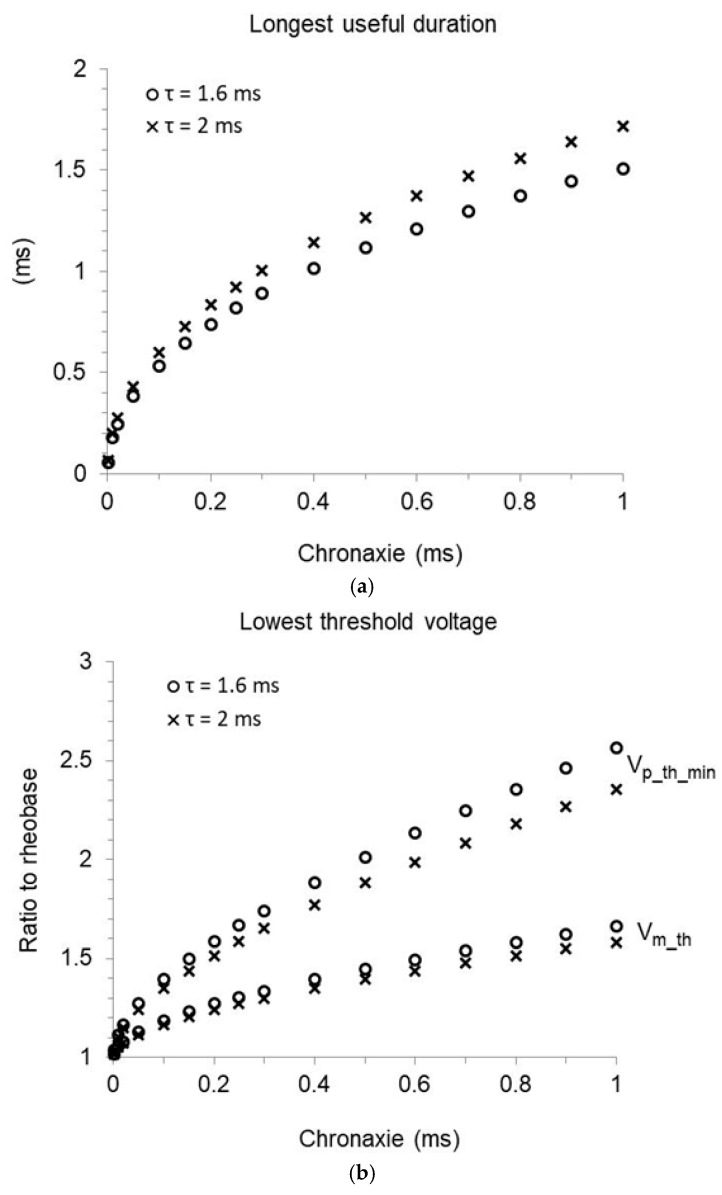
Effect of pulse decay on the threshold parameters in implants with time-constant of 1.6 or 2 ms (circles and crosses, respectively). (**a**) Longest useful pulse duration as a function of the chronaxie. Beyond this limit, the threshold start amplitude is independent of duration, and Lapicque’s law does not apply. (**b**) Lowest threshold start voltage ($V_{p\_th\_min}$) and corresponding mean voltage ($V_{m\_th}$) at the longest useful duration, normalized to the rheobase and plotted as a function of the chronaxie.

**Figure 10 bioengineering-12-00194-f010:**
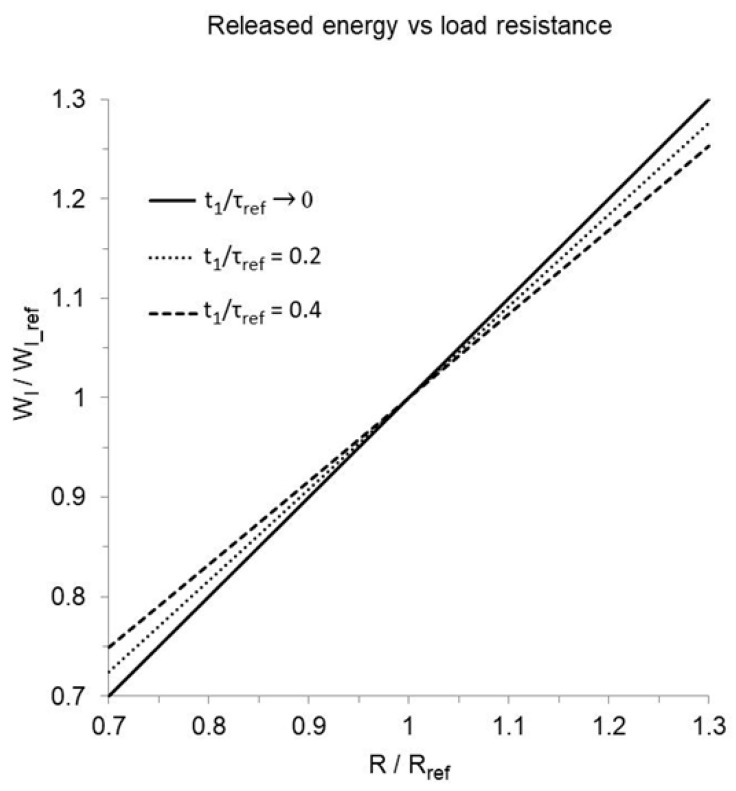
Effect of a change in load resistance ($R/R_{ref}$) on the released energy ($W_{l}/W_{l\_ref}$), calculated by assuming that pulse mean current and duration are constant, $C_{a,c}$ = 5 µF, $C_{out}$ = 4 µF, and $R_{ref}$ = 500 Ohm (from which $\tau_{ref}$ = 2 ms). The relation is virtually linear in the resistance range of ±30%, with slope decreasing for increasing ratio of pulse duration to reference time-constant ($t_{1}/\tau_{ref}$). The solid line represents a pulse duration approaching 0, featuring 1:1 correspondence between relative resistance and released energy. Dotted and dashed lines indicate, respectively, a pulse duration of 0.2 and 0.4 $t_{1}/\tau_{ref}$. Since the pacing impedance is generally lower in unipolar than bipolar stimulation, the former is better suited to pacing energy reduction.

## Data Availability

Data were produced by simulations and are contained within the article.

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
