# Peer review of "Energy Saving in Permanent Cardiac Pacing: Pulse Waveform and Charge Balancing Deserve Consideration"

_bioengineering, 2025, doi:10.3390/bioengineering12020194_

Round 1
Reviewer 1 Report
Comments and Suggestions for Authors
In my opinion methodology of presented research requires experimental verification. Novelty is thus doubtful as simulations is based on well-known relations. Conclusions are very short and general. There is not sufficient reference to state of art. Criteria of optimisation are not clearly given.
Author Response
Reply to reviewer 1
Thank you for reading the manuscript and for your comments, which were helpful in preparing the revised version. Parts added or modified are highlighted by using red characters in the submitted revised manuscript.
We perfectly agree on the point that the study requires experimental verification. Actually, our theoretical analyses was inspired by clinical experience in the optimization of energy consumption in His bundle pacing (references 29-31), where the threshold can be high and a pulse duration of 1 ms was originally recommended. Our results, as well as other published evidence (references 20,21), strongly suggest that much shorter pulses can effectively be applied in the majority of the implants. However, a systematic clinical evaluation of the issue is out of the aims of the present paper, as stated in the “Limitations”. We believe that providing a solid theoretic background can be a valuable preliminary step, which could be helpful to other clinicians and researchers.
We do claim in the “Introduction” that our simulation is based on well-known relations, leading to undisputable conclusions if the mathematical elaboration is correct (as we think it is). The novelty comes from the attempt to highlight some implications of classic concepts, which did not receive the due attention in the published literature. The relation between the starting peak amplitude of a pacing pulse generated by implantable stimulators (which is the programmable parameter known to the user) and the pulse mean amplitude (the ordinate in Lapicque’s strength-duration curve), implies the existence of a “longest useful duration”, which must never be exceeded to avoid energy waste. This rule applies to pacing either at threshold (which is important in excitability assessment) or with a safety margin (the most common condition in the clinical practice). Pacing with an additive safety margin entails the minimum energy consumption at a pulse duration substantially shorter than the chronaxie (the duration which minimizes, in contrast, the consumption induced by threshold square pulses). Bipolar pacing is often selected in permanent pacing, but the clinicians must be aware that this implies a significant increase in energy consumption, with respect to the unipolar option. We think our paper demonstrates the above in a convincing way, supporting better understanding of the excitation theory in the cardiac stimulation community, including physicians, technicians, and nurses.
The conclusions have been expanded in the revised version. We also wish to stress that each chapter features a sort of summary at its end (the message to take home). Most of these considerations are recalled in the last chapter “A practical approach to pulse optimization in cardiac pacing”, which immediately precedes the “Conclusions”.
All published papers on the application of the strength-duration curve in cardiac pacing, according to PubMed, have been cited and discussed, including 2 citations added in the revised version. The new reference 12 is particularly important, as it presents clinical observations which perfectly fit our theoretical predictions on the benefit of a pulse duration much shorter than the chronaxie, if pacing is performed with a constant safety margin. Irnich published a series of important papers on the strength-duration curve, which are the background of our work, but they deal with threshold pacing only. We cannot compare our results with previous literature, since our approach is different.
In our opinion, optimization is achieving a certain goal with minimum effort. In cardiac pacing, this general principle means ensuring effective capture of the aimed target, with the wanted safety margin and no adverse side effects, using the smallest possible energy. Care was taken to make this clear in the revised version.

Reviewer 2 Report
Comments and Suggestions for Authors
Please refer to the attachment for my comments and suggestions for Authors. Thank you!

Author Response
Reply to reviewer 2
Thank you for reading the manuscript and for your comments, which were helpful in preparing the revised version. Parts added or modified are highlighted by using red characters in the submitted revised manuscript.
We reply point-by-point to your observations in the following.
1 - Introduction
The introduction has been modified to better stress the relevance of pulse duration in energy saving. The paper shows how the duration should be managed, taking into account the actual pulse waveform and the role played by the isolation capacitors.
Two more citations have been included in the “References”, which now contain all published papers on the application of the strength-duration curve in cardiac pacing, according to PubMed. The new reference 12, published in 2024, is particularly important, as it presents clinical observations which perfectly fit our theoretical predictions on the benefit of a pulse duration much shorter than the chronaxie, if pacing is performed with an additive safety margin.
2 - Research design
2.1
Values assumed in the simulations refer to “common conditions in conventional myocardial pacing”, as now stated in the “Introduction” of the revised version. In particular, a chronaxie of about 0.4 ms is reported in most studies, and the default pulse duration is set at this value in many implantable pacemaker models. A standard 500 Ohm pacing impedance is usually applied in technical tests of stimulator performance. In some simulations, we also considered 400 Ohm as a possible alternative, which is often met in unipolar stimulation and would reduce the pulse time-constant, thus accelerating the amplitude decay. A 4 µF output capacitance is an assumption which can explain the pulse waveforms reported by many manufacturers.
All currently used implantable pacemaker models by any manufacturer show exponential decline of the output voltage during the pulse time-course. A square pulse can only be produced by non-implantable stimulators. Our simulations always refer to a pulse featuring exponential decay. Comparison with square waveforms is only presented in the text, to demonstrate the difference between the two models, clearly stating which one is described. The pulse voltage is indicated as if a squared waveform is assumed. In contrast, the symbol indicates the start voltage of a declining pulse.
In correspondence of equation 28, we now declare that “The web site https://www.derivative-calculator.net/ (David Scherfgen IT Services, Germany) was chosen as a source of reliable calculation tools”.
2.2
Rheobase and chronaxie (assumed as previously discussed) were used to build the strength-duration curve at threshold, or in the presence of a safety margin, which was converted in turn into the relevant energy-duration relation. The pulse duration entailing the lowest energy released to the load was considered as the optimal setting, i.e., that producing the aimed physiological effect with minimum energy consumption.
With square pulses, a duration equal to the chronaxie ensures threshold pacing with minimum energy expense. We demonstrate, however, that the optimal duration is shorter with decaying pulses, especially if an additive safety margin is applied.
In order to build the strength-duration curve, the pulse duration should range from less than 0.1 ms to 1 ms or more. The default pulse duration in many pacing devices is around 0.4 ms, which is supposed to be the median chronaxie in conventional implants. However, individual adaptation is always advisable. With long pulse durations, the strength-duration curve is usually flat. The constant threshold amplitude in this portion of the curve is often regarded as the rheobase, but our paper clearly demonstrates that this practice should not be applied, if threshold was determined by an implanted pacemaker with declining pulse amplitude. In this case, the lowest threshold observed is substantially higher than the rheobase.
3 – Methods
The mentioned terms are described in the chapter “Lapicque’s law of excitation” (the first after the “Introduction”).
How chronaxie and time constant values were assumed is discussed above.
4 – Results
All published papers on the application of the strength-duration curve in cardiac pacing, according to PubMed, have been cited and discussed, including 2 citations added in the revised version. The new reference 12 is particularly important, as it presents clinical observations which perfectly fit our theoretical predictions on the benefit of a pulse duration much shorter than the chronaxie, if pacing is performed with a constant safety margin. Irnich published a series of important papers on the strength-duration curve, which are the background of our work, but they deal with threshold pacing only. We cannot compare our results with previous literature, since our approach is different.
In the revised version, we added some details in the legend to some figures for better clarity. In most cases, however, full understanding requires reading the related main text, where the meaning of each simulation is reported and discussed. We made our best to integrate text and figure legends, keeping the latter concise and avoiding excessive repetition.
5 – Discussion
5.1
In the revised version, the “Conclusions” have been modified to highlight the potential implications in the clinical setting.
Our suggestions on how to manage the pulse duration are mainly addressed to clinical users (physicians, technicians, nurses) rather than manufacturers. The impact on device longevity is discussed in general terms, as it depends on the ratio between pacing consumption and total consumption, which can be different in different products. A quantitative evaluation of the energy cost of suboptimal duration setting is provided in Fig. 4b and 6b, for pacing at threshold or with additive safety margin, respectively.
Advice to the users is offered in the chapter “A practical approach to pulse optimization in cardiac pacing”, which is intended as a sort of summary and discussion of the theoretical evidence. It could be synthetized as: “keep the pulse short and select unipolar stimulation, especially is the threshold is high”.
5.2
The “Limitations” have been extended by stating that:
“The study evaluates the combined effects of safety margin, chronaxie, pulse duration and time-constant, in a range of values which apply to myocardium and conduction system pacing in normal conditions. The simulations don’t include the analysis of extreme changes in pacing impedance, due to lead failure.”
5.3
The need for clinical studies testing the practical value of the theoretical evidence has been mentioned in the revised manuscript, at the end of the chapter “A practical approach to pulse optimization in cardiac pacing”.
6 – Conclusions
In the revised version, the “Conclusions” have been modified to highlight the potential implications in the clinical setting.
7 – Language and Presentation
Long phrases have been split in the revised version. The “References” have been revised and now include 2 more citations.

Reviewer 3 Report
Comments and Suggestions for Authors
The authors presented a comprehensive study on optimizing energy consumption in permanent cardiac pacing devices by focusing on pulse waveform properties and charge balancing. The paper demonstrates the implications of pulse decay on energy efficiency and safety margins, supported by mathematical modeling and simulations. However, I have the following comments which will improve the quality of the paper:
-
The manuscript lacks a detailed comparison with existing studies addressing similar challenges in cardiac pacing, such as energy-saving algorithms, device longevity improvements, or electrode innovations. Including a section on how the proposed methodology outperforms prior works would strengthen the novelty and significance.
-
While the mathematical models and simulations are thorough, experimental results with actual pacing devices or prototypes would greatly enhance the paper's credibility. Have the authors considered implementing and testing the proposed energy optimization methods in clinical or pre-clinical scenarios?
-
The paper assumes ideal conditions for load resistance, electrode capacitance, and battery performance. It would be beneficial to discuss how real-world variability, such as tissue heterogeneity or electrode placement inaccuracies, could impact the results.
-
The choice of safety margins in the study needs elaboration. Why were specific values chosen, and how do they relate to clinical practices or regulatory requirements?
-
Some figures, such as those illustrating energy expenditure curves, could benefit from annotations or clearer legends to make them more interpretable for readers without a strong background in mathematical modeling.
The English in the manuscript is clear and professional, but some sentences are overly complex. Simplifying sentence structures and addressing minor grammatical issues would improve readability and overall quality.
Author Response
Reply to reviewer 3
Thank you for reading the manuscript and for your comments, which were helpful in preparing the revised version. Parts added or modified are highlighted by using red characters in the submitted revised manuscript.
We reply point-by-point to your observations in the following. 1.
We did not intend to cover all the aspects of energy saving in cardiac pacing, ranging from stimulator and lead design to device implant and setting. The paper is just focused on the management of the pacing system after the implant is done, especially in case of high threshold. We try to state this clearly in the revised version. Essential information on pacing lead evolution is provided in the chapters “Effects of load resistance on the pacing energy” and “Role of the electrode capacitance”, because these two parameters are relevant to our theoretical analysis.
All published papers on the application of the strength-duration curve in cardiac pacing, according to PubMed, have been cited and discussed, including 2 citations added in the revised version. The new reference 12 is particularly important, as it presents clinical observations which perfectly fit our theoretical predictions on the benefit of a pulse duration much shorter than the chronaxie, if pacing is performed with a constant safety margin. Irnich published a series of important papers on the strength-duration curve, which are the background of our work, but they deal with threshold pacing only. We cannot compare our results with previous literature, since our approach is different.
2.
Our theoretical analyses was inspired by clinical experience in the optimization of energy consumption in His bundle pacing (references 29-31), where the threshold can be high and a pulse duration of 1 ms was originally recommended. However, a systematic clinical evaluation is out of the aims of the present paper, as stated in the “Limitations”. We believe that providing a solid theoretic background can be a valuable preliminary step, which could be helpful to other clinicians and researchers.
3.
Values assumed in the simulations refer to “common conditions in conventional myocardial pacing”, as now stated in the “Introduction” of the revised version. In particular, a chronaxie of about 0.4 ms is reported in most studies, and the default pulse duration is set at this value in many implantable pacemaker models. A standard 500 Ohm pacing impedance is usually applied in technical tests of stimulator performance. In some simulations, we also considered 400 Ohm as a possible alternative, which is often met in unipolar stimulation and would reduce the pulse time-constant, thus accelerating the amplitude decay. A 4 μF output capacitance is an assumption which can explain the pulse waveforms reported by many manufacturers. Tissue heterogeneity and electrode distance with respect to the target are expected to alter rheobase and chronaxie. The effects of different chronaxie values are described in Fig. 4 and 9.
In the revised version, the “Limitations” have been extended by stating that:
“The study evaluates the combined effects of safety margin, chronaxie, pulse duration and time-constant, in a range of values which apply to myocardium and conduction system pacing in normal conditions. The simulations don’t include the analysis of extreme changes in pacing impedance, due to lead failure.”
4.
For many years, the indication of most opinion leaders was pacing at twice the threshold amplitude at the selected duration, for the sake of safety. However, this is not always possible, or convenient, if the threshold is high. On the other hand, if the threshold is very low, twice the threshold can result in too small additive margin at long durations. An additive margin is often applied in devices equipped with capture check systems, ensuring prompt amplitude increase in the event of failure. The values can range from one programming step above the threshold up to 1 V (reference 12), according to the clinical needs.
5.
In the revised version, we added some details in the legend to some figures for better clarity. In most cases, however, full understanding requires reading the related main text, where the meaning of each simulation is reported and discussed. We made our best to integrate text and figure legends, keeping the latter concise and avoiding excessive repetition.
In the revised version, we carefully checked the text and split long sentences to make them lighter and more effective.

Round 2
Reviewer 1 Report
Comments and Suggestions for Authors
The Authors wrote "We perfectly agree on the point that the study requires experimental verification" which is the keypoint referring to methodology, which according to me shoulf include practical verification. Thus, I proposed reject of the paper. It should be noted, however, that the proposed mathematical model is correct at the preliminary level, although it is not a breakthrough and has not been experimentally verified. However, if the Editor considers articles at this level of preliminary research to be acceptable and returns them for re-evaluation, then the article can be published under this assumption.
Reviewer 3 Report
Comments and Suggestions for Authors
no more comments
Comments on the Quality of English LanguageThe English could be improved to express the research more clearly.